# Effects of Parboiling on Chemical Properties, Phenolic Content and Antioxidant Capacity in Colored Landrace Rice

**DOI:** 10.3390/foods13030393

**Published:** 2024-01-25

**Authors:** Wanwipa Pinta, Chorkaew Aninbon, Phissanu Kaewtaphan, Kannika Kunyanee

**Affiliations:** 1Faculty of Natural Resources, Rajamangala University of Technology Isan, Phang Khon, Sakon Nakhon 47160, Thailand; wanwipanoi@gmail.com; 2Faculty of Agricultural Technology, King Mongkut’s Institute of Technology Ladkrabang, Bangkok 10520, Thailand; phissanu.ka@kmitl.ac.th; 3Faculty of Food Industry, King Mongkut’s Institute of Technology Ladkrabang, Bangkok 10520, Thailand; kannika.ku@kmitl.ac.th

**Keywords:** brown rice, native rice, parboiling, streaming, phenols, DPPH, chemical compositions

## Abstract

Parboiling influences chemical compositions in rice grains. The objectives of this study were to evaluate the change in chemical content, total phenolic content and antioxidant capacity of landrace rice genotypes under parboiling conditions and to identify the genotypes suitable for production of parboiled rice. Landrace rice varieties used in this study consisted of Glam Feang, Glam Tonkeaw, Kawgum, Glam Luem Phua, Medmakham, Deang Sakonnakhon, Sang Yod, Kawniewd-eang, Mali Deang, KDML105 and RD6. Parboiling reduced fiber content, total phenolic content and 2,2-diphenyl-1-picrylhydrazyl (DPPH) radical-scavenging activity in rice grains. Fiber contents were 1.46% in brown rice (unpolished rice) and 1.40% in parboiled rice (24 h of soaking and 48 h of incubation). Total phenolic contents were 205.67 mg/100 g seed in brown rice and 35.34 mg/100 g seed in parboiled rice. Antioxidant capacity (DPPH) reduced from 68.45% in brown rice to 26.23% in parboiled rice. Ash content and protein content were not significantly affected by the parboiling process. Medmakham cv. had the highest total phenolic content and antioxidant capacity in brown rice and parboiled rice. Gum Leamphea cv. and Medmakham cv. were the best genotypes for ash content, protein content, total phenolic content and antioxidant capacity (DPPH) in brown rice and parboiled rice. Glam Feang cv. had the highest protein contents in brown rice and parboiled rice although it had low total phenolic content and antioxidant capacity. Cluster analysis further showed variation among genotypes, revealing distinct groupings in brown rice and parboiled rice based on chemical properties, phenolic content and antioxidant capacity. This research significantly contributes to a better understanding on how parboiling affects rice compositions and nutritional values. It emphasizes the importance of nuanced comprehension of how different rice varieties respond to parboiling, aiding informed decisions in rice processing and selection to meet specific nutritional needs.

## 1. Introduction

Thailand is rich in rice diversity and has been a leading rice exporter for decades. Rice diversity in Thailand has been preserved in rice cultivation. However, the diversity has gradually reduced because the landrace varieties have been replaced by newly improved varieties. Rice diversity is beneficial for maintaining parent stock in rice breeding and can also serve as an incentive for farmers to cultivate various varieties. Maintaining rice diversity in rice cultivation is another way to preserve the landrace varieties. Although the majority of rice landraces have been preserved in gene banks, they are at risk of depletion.

Moreover, these different rice varieties exhibit varying nutritional profiles. Longvah et al. [1] reported that landrace rice was a good source of nutrients. Rice grains consisted of lipid, dietary fiber, ash, carbohydrates and protein [2,3]. In addition, rice grains also contain significant amounts of antioxidants in the phenol group, such as phenolic acid, ferulic acid, flavonoids [4,5] and anthocyanin [6,7]. These phytochemicals are beneficial to human health because they are important antioxidants.

Consumers are becoming increasingly concerned about health problems from consuming harmful food [8], and the demand for health food products is increasing. Colored rice contains a higher phenol content and greater antioxidant capacity than non-colored rice [9]. Colored rice has the potential to become a popular choice among health-conscious consumers.

Parboiled rice is a product from rice that is obtained by soaking paddy rice in water until the moisture content is approximately 30–40%, then steaming or boiling until cooked. It is then dried to an appropriate humidity level and then milled to remove the hull [10]. Parboiling improves the milling quality, makes the rice less broken and improves the nutritional value of the rice [11] because the nutrients from the peel seep into the grain during soaking and steaming. Germinated parboiled rice involves the germination process of rice grains. The germination process increases the nutrients in rice grains such as γ-amino butyric acid (GABA), fiber, vitamins and phenolic acids [12]. Addition, parboiling rice can increase phenol content such as total free phenolic content, p-coumaric acid and bound vanillic acid [13].

Although parboiled rice is not consumed by local consumers, it has been an export product of Thailand for a long time just like milled rice [14]. Good-quality parboiled rice in Thailand still has a lot of room for development. Therefore, parboiled rice is a product that should be promoted and developed, especially colored parboiled rice for a specific market segment for health care in Thailand.

The information on germinated parboiled colored rice is still incomplete, especially for a long germination time of 48 h. There are several changes after parboiling. The changes are mainly beneficial for conserving rice quality for long-term storage although there are some unfavorable effects. Extending incubation time to 48 h can lead to an increase in useful phytochemicals, and, therefore, we selected 48 h of incubation time for this study. The assumption underlying the research project is that the landrace rice genotypes respond differently to the parboiling process in terms of chemical compositions, total phenolic content and antioxidant capacity.

The objectives of this study were to evaluate the change in chemical content, total phenolic content and antioxidant capacity of landrace rice genotypes under parboiling conditions and to identify the genotypes suitable for production of parboiled rice. The information on the genotypes with high contents of phytochemicals and antioxidants could be useful for direct use in rice production and rice breeding.

## 2. Materials and Methods

### 2.1. Experimental Design

The experiment was conducted during July to December 2021 at a farmer’s field in the province of Sakon Nakhon (17.1664° N, 104.1486° E). Eleven rice varieties were planted in a randomized complete block design with three replications. Rice varieties used in this study consisted of five black landraces, four red landraces and two white commercial varieties. The nine landrace rice varieties used in this study were selected based on high yield. Glam Feang cv., Glam Tonkeaw cv., Kawgum cv., Glam Luem Phua cv. and Medmakham cv. are black landraces. Deang Sakonnakhon cv., Sang Yod cv., Kawniewdeang cv. and Mali Deang cv. are red landraces, and KDML105 cv. and RD6 cv. are white commercial varieties. The soil was plowed twice and puddled before planting. Seedlings were transplanted at the age of 25 days after sowing at the rate of one plant per hill. Plot size was 2 × 2 m, and the seedlings were transplanted at a spacing of 25 × 25 cm between rows and between plants within rows. Compound fertilizer (15–15–15 of N–P–K) at the rate of 156.25 kg ha^−1^ was applied in two parts at 30 and 60 days after transplanting.

### 2.2. Brown Rice and Parboiled Rice Preparation

Paddy rice samples of 11 varieties were separated into two groups. The first group was dehulled by machine to obtain brown rice. The paddy rice samples of the second group were used for making parboiled rice, which were soaked in distillswater for 24 h and incubated at room temperature for 48 h until roots germinated at 0.5 cm. Then, the germinated rice samples were steamed at a temperature of 95–97 °C and sun-dried for 2 days until the moisture content was 11 ± 1%. After that, all rice samples were dehulled by a machine. Finally, the grain samples of brown rice and parboiled rice were ground into fine powder and stored in plastic bags at 4 °C for further analysis. Single samples without replicates were analyzed for all parameters.

### 2.3. Lipid Content Analysis

Lipid content was analyzed based on the Association of Official Agricultural Chemists (AOAC) 922.06 [15]. Two grams of ground rice samples was weighed into a thimble tube and put into a beaker. Lipid was extracted with 40 mL of petroleum ether. The samples were extracted for 1 h and the solvents were evaporated. The beakers were heated at 130 °C for about 1 h, cooled in a desiccator and weighed.

### 2.4. Ash Content Analysis

The ground rice samples (2 g) were added to a crucible and burned on a hot plate for 30 min, followed by burning in a muffle furnace at 550 °C for 6 h [15]. The samples were cooled until the temperature dropped below 200 °C. The samples were taken from the kiln, put in a desiccator and cooled for 30 min. The samples were weighed in the desiccator, and ash weight was recorded.

### 2.5. Fiber Content Analysis

Fiber content was measured based on the method recommended by AOAC 985.29 [15]. Approximately 1 g of Celite 545 was loaded into the crucible, and 1 g of ground rice sample was loaded into the crucible. In case the sample had lipid content higher than 1%, the sample was defatted before subsequent digestion. Twenty-five milliliters of acetone was added to the sample. The sample was soaked for 10 min and filtered.

Then, 1.25% of warm sulfuric acid was added to a 150 mL volumetric flask, and 2–3 drops of n-Octano were added to prevent foam formation. The sample was heated until boiling. Heat was reduced after boiling. The sample was boiled for 30 min and filtered.

The filter was washed with hot water 3–5 times, and 30 mL of the sample was evaporated until dry. The dry sample was added with 25 mL of acetone, soaked for 10 min and filtered three times. The crucible was fired at 525 °C for 5 h, cooled in a desiccator and weighed.

### 2.6. Protein Content Analysis

Two grams of each ground rice sample was loaded into tubes for protein digestion. Then, the tubes were filled with copper sulfate and two seeds of crystal balls. After that, 20 mL of 95% concentrated sulfuric acid was added to the tubes, and the tubes were placed into the protein digester for protein digestion.

Digestion was performed at temperatures from 150–400 °C for 4–5 h. Digestion reaction was continued until a clear green solution was obtained. Then, the samples were set aside to reduce the temperature until the temperature dropped at room temperature. After cooling, 20 mL of distilled water was added to the samples followed by 80 mL of 40% NaOH. The samples were titrated with a solution of 0.1 N HCl and the volume of 0.1 N HCl used in the titration was recorded [15].

### 2.7. Total Phenolic Content and Antioxidant Capacity Analysis

Total phenolic content was measured using the Folin–Ciocalteu reaction [7]. Two grams of each ground rice sample was extracted with 10 mL of methanol. Then, the samples were centrifuged at 5000 rpm for 10 min and filtered through no.4 Whatman paper. The extracted samples (50 μL) were added to 3.0 mL of distilled water. After that, 250 μL of Folin–Ciocalteu reagent was added to the tubes, and 20% sodium carbonate solution at 750 μL was added to the samples followed by adding 950 μL of distilled water. The solutions were measured at a wavelength of 765 nm using a spectrophotometer. The amount of total phenolic content was expressed as milligrams of gallic acid equivalent (GAE) per 100 g of dry seeds (mg GAE/100 g).

DPPH radical-scavenging activity was analyzed based on Kunnam et al. [7]. Briefly, 200 μL of DPPH solution and 2.8 mL of methanol were added to the extracted samples (100 μL). The mixed samples were shaken well and stored in the dark for 30 min. Measurement of the absorbance was carried out at a wavelength of 515 nm. The absorbance was used to calculate the removal percentage of DPPH free radicals.

### 2.8. Data Analysis

Analysis of variance (ANOVA) was performed based on a 2 × 11 factorial experiment in a randomized complete block design, in which type of rice grain (brown rice and parboiled rice) and rice genotype were factors. Duncan’s multiple range test (DMRT) was used for mean comparison at 0.05 and 0.01 probability levels. A *t*-test was calculated to determine the difference between the same genotype in different grain types. Cluster analysis was conducted using the SAS program (SAS Institute Inc., Cary, NC, USA). The Pearson correlation coefficient was used to analyze the relationships among all traits.

## 3. Results

### 3.1. Analysis of Variance

The types of rice grain (brown rice and parboiled rice) were significantly different (*p* ≤ 0.05 and 0.01) for fiber content, fat content, total phenolic content and antioxidant capacity, and they were not significantly different for ash content and protein content (Table 1). Rice varieties were significantly different (*p* ≤ 0.01) for all traits, and the interactions between type and variety were significant (*p* ≤ 0.01) for most traits except for protein content. The results indicated that rice genotypes respond similarly to parboiling regarding protein content. Significant interactions in this study indicated that the genotypes performed well in brown rice and parboiled rice may not be the same. Therefore, it is important to identify the best genotypes for each character in both grain types.

### 3.2. Means of Brown Rice, Parboiled Rice and 11 Rice Varieties

Brown rice had significantly higher fiber content, total phenolic content and DPPH radical-scavenging activity than parboiled rice, whereas parboiled rice had significantly higher lipid content than brown rice (Table 2). Brown rice and parboiled rice were similar for ash content and protein content. It is interesting to note here that parboiling increased lipid content from 2.08 to 2.33%, and fiber content was reduced from 1.46 to 1.40%. High reductions after parboiling were found for total phenolic content and DPPH. Total phenolic content was reduced from 205.67 mg/100 g seed in brown rice to 35.34 mg/100 g seed in parboiled rice, and DPPH was reduced from 68.45% in brown rice to 26.23% in parboiled rice.

Rice genotypes were significantly different for ash content, lipid content, fiber content, protein content, total phenolic content and DPPH radical. Ash contents ranged from 0.89% in Kawgum cv. to 1.54% in Glam Luem Phua cv. Lipid contents ranged between 1.93% in RD6 cv. and 2.78% in Sang Yod cv. Fiber contents ranged between 1.06% in RD6 cv. and 2.13% in Medmakham cv.

Rice genotypes had rather high protein contents, ranging from 7.87% in KDML 105 cv. to 12.60% in Glam Feang cv., and they had a wide range of total phenolic contents (8.07 mg/100 g seed in RD6 cv. and 205.46 mg/100 g seed in Medmakham cv.). Rice genotypes also had a wide range of DPPH values, ranging from 6.63% in KDML 105 cv. to 68.63% in Glam Luem Phua cv. Glam Luem Phua cv. was the most interesting genotype for the highest DPPH (68.63%) and ash content (1.54%), whereas Medmakham cv. was the most interesting genotype for the highest total phenolic content (205.46 mg/100 g seed) and DPPH (65.76%).

### 3.3. Comparison between Brown Rice and Parboiled Rice

The interactions between rice genotype and grain type were significant for most traits, indicating the differential responses of rice genotypes to parboiling. Therefore, brown rice and parboiled rice were compared for each genotype.

#### 3.3.1. Ash Content

The effect of interaction between grain type and genotype was rather high for ash content (Table 3). Glam Feang cv. and Kawniewdeang cv. were not different between the two grain types. Glam Luem Phua cv., Medmakham cv., Deang Sakonnakhon cv., Mali Deang cv., KDML105 cv. and RD6 cv. had higher ash contents in parboiled rice, whereas Galm Tonkeaw cv., Kawgum cv.,and Sang Yod cv. had higher ash contents in brown rice. Sang Yod had the highest ash content in brown rice (1.84%), and Glam Luem Phua cv. had the highest ash content in parboiled rice (1.68%).

#### 3.3.2. Lipid Content

Six rice genotypes including Glam Tonkeaw cv., Kawgum cv., Glam Luem Phua cv., Medmakham cv., Deang Sakonnakhon cv. and Kawniewdeang cv. did not showed significant differences between brown rice and parboiled rice for lipid content (Table 3). Glam Feang cv., Mali Deang cv. and RD6 cv. showed higher lipid contents (2.35, 2.55 and 2.73%, respectively) in parboiled rice, and Sang Yod cv. and KDML105 cv. showed higher lipid contents (3.34 and 2.45%, respectively) in brown rice. Sang Yod cv. also had the highest lipid content in brown rice, whereas RD6 cv. had the highest lipid content in parboiled rice.

#### 3.3.3. Fiber Content

Six genotypes, Glam Feang cv., Kawgum cv., Glam Luem Phua cv., Medmakham cv., Sang Yod cv. and Kawniewdeang cv., had significantly reduced fiber contents after parboiling, and two genotypes, Mali Deang cv. and RD6 cv., had significantly increased fiber contents after parboiling, whereas three genotypes, Glam Tonkeaw cv., Deang Sakonnakhon cv. and KDML105 cv., had similar fiber contents in brown rice and parboiled rice (Table 3). Medmakham cv. was the best genotype for fiber content in brown rice and parboiled rice, and Mali Deang cv. was also a good genotype for fiber content in parboiled rice.

#### 3.3.4. Protein Content

The effect of parboiling on protein content was low, and nine genotypes, Glam Feang cv., Glam Tonkeaw cv., Glam Luem Phua cv., Medmakham cv., Deang Sakonnakhon cv., Sang Yod cv., Kawniewdeang cv., Mali Deang cv. and RD6 cv., had similar protein contents in brown rice and parboiled rice (Table 4). Kawgum cv. had higher protein content in parboiled rice (10.47%), and KDML105 cv. has higher protein content in brown rice (8.77%). The highest protein contents were found in Glam Feang cv. in brown rice (12.72%) and parboiled rice (12.49%).

#### 3.3.5. Total Phenolic Content (TPC)

Parboiling significantly reduced total phenolic content in most rice genotypes except two for KDML105 cv. and RD6 cv., which showed similar total phenolic contents in brown rice and parboiled rice (Table 4). Medmakham cv. had the highest total phenolic content in brown rice (358.25 mg/100 g seed). Glam Luem Phua cv. had the highest total phenolic content in parboiled rice (79.29 mg/100 g seed).

#### 3.3.6. Antioxidant Capacity (DPPH)

Parboiling significantly reduced DPPH in most rice genotypes except KDML105 cv., which showed similar total phenolic contents in brown rice and parboiled rice (Table 4). The genotypes with black grain and the genotypes with red grain had much higher DPPH than the genotypes with white grain, and six genotypes, Glam Luem Phua cv., Medmakham cv., Deang Sakonnakhon cv., Sang Yod cv., Kawniewdeang cv. and Mali Deang cv., had the highest DPPH values in brown rice, ranging from 85.24 to 87.47%, which were not significantly different. Glam Luem Phua cv. had the highest DPPH value (52.01%) in parboiled rice.

### 3.4. Relationships among Traits

Protein content was not significantly correlated with other parameters including fiber content, ash content, lipid content, total phenolic content and DPPH (Table 5). Ash content, fiber content and lipid content were significantly correlated with each other, and the correlation coefficients (r) were moderate, ranging from 0.2611 * to 0.4357 **. Fiber content also significantly correlated with phenolic content (r = 0.3409 **) and DPPH (r = 0.3391 *), whereas lipid content had moderate correlation with DPPH (r = 0.4357 **). Although the correlation coefficient between ash content and DPPH was not significant, it was positive (r = 0.2308). The highest correlation coefficient was found between total phenolic content and DPPH (r = 0.9470 **).

### 3.5. Cluster Analysis

Cluster analysis classified 11 rice genotypes into distinct groups based on ash content, fiber content, lipid content, protein content, total phenolic content and antioxidant capacity in brown rice (Figure 1) and parboiled rice (Figure 2). Based on the phytochemicals and antioxidant capacity in brown rice, the dendrogram classified 11 rice genotypes into four groups (Figure 1).

Group 1 had Glam Feang cv. as an only one member of this group andthis group had high ash content, high protein content, low total phenolic content and low DPPH. KDML105 cv. and RD6 cv. were in group 2, and this group had low ash content, intermediate lipid content, low fiber content, low protein content, low total phenolic content and low DPPH. Group 3 had four genotypes consisting of Glam Tonkeaw cv., Sang Yod cv., Kawniewdeang cv. and Kawgum cv. This group was intermediate for ash content, lipid content, fiber content and total phenolic content, but it was relatively high for protein content and DPPH. Group 4 had four genotypes consisting of Glam Luem Phua cv., Deang Sakonnakhon cv., Mali Deang cv. and Medmakham cv. This group was characterized by low ash content, low lipid content, low fiber content, low to intermediate protein content, intermediate to high total phenolic content and intermediate to high DPPH.

Based on the phytochemicals and antioxidant capacity in parboiled rice, the dendrogram classified 11 rice genotypes into four groups (Figure 2). Group 1 had four genotypes consisting of Glam Feang cv., Sang Yod cv., Deang Sakonnakhon cv., and Mali Deang cv. The characteristics of this group were low to intermediate ash content, intermediate to high lipid content, intermediate to high fiber content, high protein content, low total phenolic content and low DPPH. Group 2 had two genotypes, KDML105 cv. and RD6 cv. This group had intermediate ash content, low to high lipid content, low fiber content, low protein content, low total phenolic content and low DPPH. Group 3 had four genotypes, Glam Tonkeaw cv., Kawgum cv., Kawniewdeang cv. and Medmakham cv. This group had intermediate ash content, intermediate lipid content, low fiber content, intermediate protein content, intermediate total phenolic content and intermediate DPPH. Glam Luem Phua cv. was in group 4 with high ash content, intermediate lipid content, low fiber content, high protein content, high total phenolic content and high DPPH.

## 4. Discussion

### 4.1. Ash Content

Ash content is an important food component that should be labeled in food products. In rice, studies reported that ash contents were in the ranges of 0.8 to 1.3% in brown rice [16], 1.48 to 1.98% in dehulled rice [17] and 0.56 to 1.05% in brown rice [18]. Ash contents in this study were in a range from 0.89% to 1.54%, which was the same range as in previous reports.

In this study, parboiling did not significantly affect ash content. In a previous study, ash contents in parboiled rice of different varieties ranged between 0.45 and 2.46% [19]. This range was wider than in this study. Alexandre et al. [20] reported that ash content in parboiled rice was dependent on soaking time, and six hours of soaking in a water bath at 80 °C reduced ash content, whereas four hours of soaking increased ash content. Therefore, different results in different studies could be in part due to different soaking conditions. Variation in ash content also depended on rice varieties. In this study, Glam Luem Phua cv. had the highest ash content (1.68%) in parboiled rice, whereas Sang Yod cv. had the highest ash content (1.84%) in brown rice.

### 4.2. Lipid Content

Lipid is an important food component, and rice bran oil is considered as healthy cooking oil. Rice is not an oil crop, and it has low contents of lipid ranging from 0.21 to 1.29% in brown rice [21]. Lipid is more concentrated in brown rice than milled rice [22]. In our study, the range of lipid contents of rice varieties averaged across two grain types (brown and parboiled) was between 1.93% and 2.78%. The range of lipid contents in this study was similar to those in previous reports.

In this study, parboiling significantly increased lipid content. Alexandre et al. [20] reported that parboiling increased lipid content in rice. Sareepuang et al. [23] reported a significant increase in lipid content as a result of the parboiling process. The lipid bodies and spherosomes of non-starch lipids are broken down and lipid is liberated from the kernel’s surface [24]. As the lipid diffuses to the surface, the bran of parboiled rice becomes greasy. Brown rice kernels after milling had lower lipids than parboiled rice kernels [19]. The increase in lipid content after the parboiling process may be attributed to the biosynthesis of new compounds during treatments [25].

However, some rice genotypes did not increase in lipid content after parboiling because of significant genotype × grain type interactions. Some genotypes had reduced lipid content after parboiling. According to Houssou et al. [19], the mean overall lipid content is low in parboiled rice as compared with non-parboiled rice of the same variety: 0.29% and 0.85%, respectively. Kata et al. [26] found that parboiling reduced some nutritional parameters like crude protein content and crude fat content and increased other parameters like ash and crude fiber content. There might be other factors affecting lipid content during the parboiling process and causing the discrepancy of the results among different studies. Sang Yod cv. had the highest lipid content in brown rice, and RD6 cv. had the highest lipid content in parboiled rice.

### 4.3. Fiber Content

Dietary fiber is important component of food, and it plays a role in the digestive system. Dietary fiber does not assimilate into the blood system, and foods with balanced dietary fiber are useful to health. Dietary fiber contents in previous reports were in ranges from ∼6 to 10% [27] and 5.5 to 8.0% [28]. In this study, fiber contents ranged between 1.06% in RD6 cv. and 2.13% in Medmakham cv. The results were similar to those reported in a previous study. Higher fiber contents in previous studies could have been caused by including resistant starch in the fiber contents.

In this study, parboiling significantly reduced fiber content. Savitha and Singh [27] found that dietary fiber in rice was low about 1% after parboiling. Joseph et al. [29] also found that parboiling reduced crude fiber content from 2.9 to 2.1% (*p* < 0.05) in Ghana rice. David et al. [30] found that crude fiber contents of parboiled rice was higher compared to that of non-parboiled rice.

The results suggested that the variability in fiber contents after the parboiling process could be influenced by several factors including rice variety [27], genetic variability [31], processing conditions and leaching of soluble fiber during rice soaking [32]. Medmakham cv. and Sang Yod cv. had the highest fiber contents in brown rice and parboiled rice.

### 4.4. Protein Content

Rice generally has protein as a second dominant component after carbohydrate, ranging from 6 to 7% [33]. The protein contents of different varieties were in the range from 5% to 13% [34]. The protein contents in rice seeds were approximately 7–9% [3,20]. The protein contents in this study were rather high, ranging from 7.87% to 12.60%, and the protein contents in this study were in the ranges reported in a previous study. However, high protein contents cause poor palatability, and high-grade rice should have protein content lower than 6%. High-protein rice is suitable for processing, and it can diversify rice products. The genotypes with high protein contents in this study can be used as a good source of protein for plant-based protein food products.

In this study, parboiling did not significantly affect protein content. Protein plays a crucial role in the texture of cooked rice due to the formation of a complex with starch, which impairs the swelling of starch granules. Starch granule swelling affects both viscosity intensity and the rate of starch gelatinization [32]. The structural alteration of parboiled rice is associated with changes in protein content during heat–moisture treatment and the subsequent drying process. Despite the rupture of protein bodies in rice kernels during the steaming process, the overall protein content of parboiled rice remains unchanged [24,35]. Although parboiling changes protein contents, the differences were not significant [30]. The results in this study were in agreement with those reported previously. Glam Feang cv. was identified as the best genotype for high protein content in brown rice and parboiled rice.

### 4.5. Total Phenolic Content (TPC)

According to Shen et al. [36], phenolic contents in whole rice grains of different grain colors, sizes and weights ranged from 108.1 to 1244.9 mg GAE/100 g, and phenolic contents were higher in colored rice than in white rice. Total phenolic content in rice endosperm was 56.95 mg GAE/100 g, whereas total phenolic content in rice bran was 596.3 mg GAE/100 g [37]. In Japonica brown rice, the total phenolic contents were in the range of 241.98–296.76 GAE mg/100 g [5]. In this study, total phenolic contents were in a range of 8.07 mg/100 g seed to 205.46 mg/100 g. The results in this study were much lower than those in previous studies.

In this study, parboiling greatly reduced total phenolic content. Contrasting results have been reported in previous studies. Parboiled rice showed higher phenolic contents compared to non-parboiled rice [38,39]. Total phenolic content increased by 23% and 20% in parboiled BRRI 28 and Katari Bhog rice extracts, respectively, due to the cooking process [40]. However, many studies reported the reduction in total phenolic content after parboiling. According to Widyasaputra et al. [41], total phenolic content of normal black rice was 12.22 gGAE/100 g. After the parboiling process, the total phenolic content was decreased to 7.67 gGAE/100 g in process A and 9.52 gGAE/100 g in process B. A similar reduction in total phenolic content after parboiling has been reported [42].

Differences in the results among different studies could be mainly due to rice varieties used, milling levels and parboiling processes. The reduction in phenolic content in this study could be due to a longer incubation time of 48 h for germination. A shorter incubation time or merely imbibition without incubation might reduce loss of phenolic content in colored rice. Moreover, some rice varieties used in this study are colored rice, which are already high in phenolic content before parboiling.

Surha and Koh [43] reported that cooking methods had a negative effect on anthocyanins, total phenolic content and antioxidant properties. The greatest reduction in total phenolic content was caused by roasting followed by frying, steaming and boiling, and steaming is an important step in the parboiling process. Therefore, parboiling in this study resulted in a reduction in total phenolic content and antioxidant capacity. Moreover, Hu et al. [13] found that total phenolic content was higher in red rice (15.3 mg GAE/100 g) compared to germinated red rice (7.1 mg GAE/100 g) and germinated red rice parboiled for 15 min (12.6 GAE/100 g). Parboiling reduced the total soluble phenolic compound concentration in the grains due to the loss of part of them in the processing water, thermal decomposition and, possibly, interaction with other components [42]. Medmakham cv. had the highest total phenolic content in brown rice, and Glam Luem Phua cv. had the highest total phenolic content in parboiled rice.

### 4.6. Antioxidant Capacity (DPPH)

The ranges of DPPH in rice in previous investigations were 10.7 to 87.9% in rice bran [44], 39.5 to 96.3% in rice bran with different extraction methods [45] and 69.22 to 81.25% in brown rice [7]. In this study, the values of DPPH were in a range between 6.63% and 68.63%. The range in this study was in the range of previous studies.

In this study, parboiling greatly reduced DPPH. The reductions in DPPH were highest in the genotypes with high DPPH such as Medmakham cv., Deang Sakonnakhon cv., Sang Yod cv., Kawniewdeang cv. and Mali Deang cv., and the reductions were lowest in the genotypes with low DPPH such as KDML105 cv. and RD6 cv. KDML105 cv. and RD6 cv. are white rice and the others are colored rice.

In a previous study, the parboiling process showed greater total phenolic content (TPC) and 2,2-diphenyl-1-picrylhydrazyl (DPPH) radical-scavenging activity when compared to non-parboiling treatments [46]. Parboiled germinated brown rice significantly exhibited the highest total phenolic content and DPPH radical inhibition compared to brown rice and white rice [47]. However, parboiling reduced the total phenolic content and antioxidant activity due to the loss of part of them in the processing water, thermal decomposition and, possibly, interaction with other components [42]. There might be many factors affecting the quality of parboiled rice, and these factors cause differences among the results of different studies. Glam Luem Phua cv., Medmakham cv., Deang Sakonnakhon cv., Sang Yod cv., Kawniewdeang cv. and Mali Deang cv. had the highest DPPH in brown rice, and Glam Luem Phua cv. also had the highest DPPH in parboiled rice.

Colored rice may not be suitable for parboiling in terms of reductions in phenolic content and DPPH. However, parboiling is beneficial in terms of long shelf life of export products. The results in this study provide useful information for the parboiled rice industry and rice consumers.

### 4.7. Relationships among Traits

In this study, ash, fiber and lipid were associated with each other, and ash and fiber were also positively associated with DPPH, whereas protein was not. Furthermore, total phenolic content was positively correlated with DPPH. Total phenolic content greatly contributed to DPPH followed by lipid and fiber.

It is well known that phenolic acids are important antioxidant compounds, and they are a major contributor to antioxidant activity [48]. Kuar et al. [49] also found the correlation between total phenolic content and antioxidant capacity measured by the DPPH method (r = 0.904 **). This information is important for rice breeding aiming to improve grain nutrient quality and quality labeling of rice products.

### 4.8. Cluster Analysis

Clustering of rice genotypes into distinct groups can be used for effective utilization of these rice genotypes for breeding and immediate use in rice production. Group 2 in brown rice and group 2 in parboiled rice were identical, consisting of KDML105 cv. and RD6 cv. KDML105 cv. and RD6 cv. are commercial varieties, and they are the most popular in terms of production area. KDML105 cv. is a fragrant, non-glutinous rice, while RD6 cv. is fragrant glutinous rice. Both of them are grown under rainfed conditions. The new version of RD6 cv. was released recently, and this version is non-photoperiod sensitive.

Nine landraces could be classified into three groups for each grain type. They formed new groups after parboiling, which were different from the old groups. This could have been caused by the effects of genotype by grain type interaction. Glam Feang cv. was still in group 1 after parboiling, and this group had new members consisting of Sang Yod cv., Deang Sakonnakhon cv. and Mali Deang cv.

Glam Tonkeaw cv., Kawgum cv. and Kawniewdeang cv. were still in group 3, whereas Sang Yod cv. was replaced by Medmakham cv. In group 4, Glam Luem Phua cv. was still in the same group, and other group members moved to other groups.

## 5. Conclusions

The results in this study added more information on how rice compositions and nutritional values are interconnected after parboiling of germinated rice. In this study, rice varieties showed variations in ash content, lipid content, fiber content, protein content, total phenolic content and antioxidant capacity, and these varieties responded differently to parboiling. Therefore, selection of rice varieties for making parboiled rice with unique qualities is important. Parboiling did not alter protein content in rice. Glam Feang cv. is an interesting genotype for its high protein content. Remarkably, Glam Luem Phua cv. and Medmakham cv. exhibited rich sources of ash, protein, total phenolic content and antioxidants in both brown and parboiled rice.

Moreover, our study examined the correlations among chemical properties and antioxidants, offering valuable insights into the factors influencing rice quality. Cluster analysis classified the 11 rice varieties into distinct groups based on chemical properties, phenolic content and antioxidant capacity pre- and post-parboiling.

In conclusion, our research significantly contributes to a better understanding on how parboiling affects rice compositions and nutritional values. It also provides a better understanding on how different rice varieties respond to the parboiling method proposed in this study. The results in this study can help the rice-parboiling industry to select suitable rice varieties for parboiling to meet specific nutritional needs. Future studies could explore the underlying mechanisms driving observed variations and their potential implications for human health and dietary considerations.

## Figures and Tables

**Figure 1 foods-13-00393-f001:**
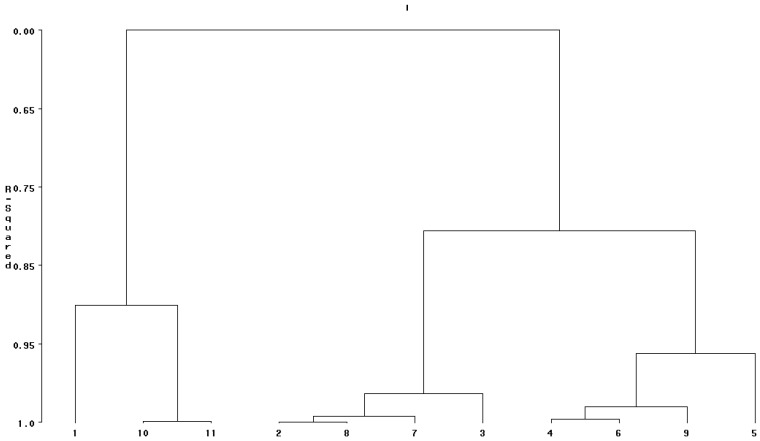
Dendrogram of 11 brown rice genotypes based on ash content, fiber content, lipid content, protein content, total phenolic content and DPPH. 1 = Glam Feang cv., 2 = Glam Tonkeaw cv., 3 = Kawgum cv., 4 = Glam Luem Phua cv., 5 = Medmakham cv., 6 = Deang Sakonnakhon cv., 7 = Sang Yod cv., 8 = Kawniewdeng cv., 9 = Mali Deng cv., 10 = KDML105 cv. and 11 = RD6 cv.

**Figure 2 foods-13-00393-f002:**
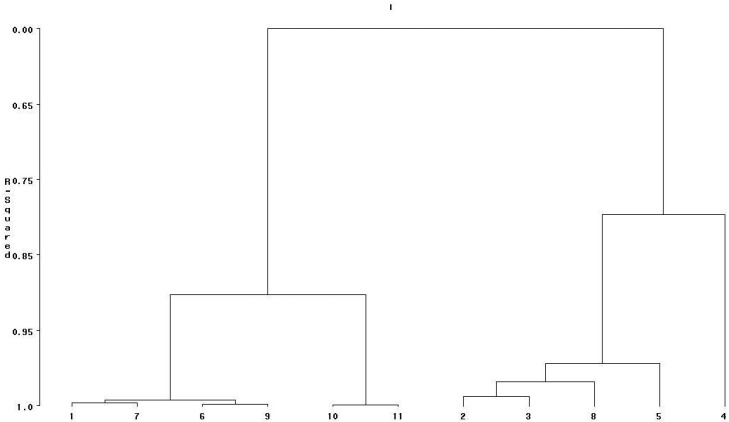
Dendrogram of 11 genotypes of parboiled rice based on ash content, fiber content, fat content, protein content, total phenolic content and DPPH. 1 = Glam Feang cv., 2 = Glam Tonkeaw cv., 3 = Kawgum cv., 4 = Glam Luem Phua cv., 5 = Medmakham cv., 6 = Deang Sakonnakhon cv., 7 = Sang Yod cv., 8 = Kawniewdeng cv., 9 = Mali Deng cv., 10 = KDML105 cv. and 11 = RD6 cv.

**Table 1 foods-13-00393-t001:** Mean squares for ash content, lipid content, fiber content, protein content, total phenolic content and antioxidant capacity (DPPH) of 11 rice varieties from combined analysis of brown rice and parboiled rice.

Source of Variation	df	Ash Content	Lipid Content	Fiber Content	Protein Content	Phenolic Content	DPPH
Rep	2	0.008	0.05	0.004	4.99	17	5.1
Type	1	0.016 ^ns^	1.06 **	0.059 *	0.74 ^ns^	478,688 **	29,416.3 **
Var	10	0.216 **	0.33 **	0.548 **	10.46 **	25,632 **	2786.9 **
Type × Var	10	0.301 **	0.71 **	0.334 **	1.42 ^ns^	15,413 **	875.2 **
Error	42	0.013	0.07	0.011	2.25	41	6.5
Total	65						
CV (%)		8.80	12.10	7.54	15.48	5.34	5.38

^ns^, *, ** = non-significant and significant at 0.05 and 0.01 probability levels, respectively.

**Table 2 foods-13-00393-t002:** Means for ash content (%), lipid content (%), fiber content (%), protein content (%), total phenolic content (mg/100 g seed) and DPPH (%) of brown rice, parboiled rice and 11 rice varieties.

	Ash Content (%)	Lipid Content (%)	Fiber Content (%)	Protein Content (%)	Total Phenolic Content (mg/100 g Seed)	DPPH (%) ^1^
Type						
Brown rice	1.29	2.08 ^b^	1.46 ^a^	9.81	205.67 ^a^	68.45 ^a^
Parboiled rice	1.32	2.33 ^a^	1.40 ^b^	9.59	35.34 ^b^	26.23 ^b^
Variety						
Glam Feang	1.34 ^c,d,e^	2.09 ^b,c,d^	1.36 ^d,e^	12.60 ^a^	68.12 ^f^	32.27 ^e^
Glam Tonkeaw	1.49 ^a,b^	2.26 ^b,c^	1.26 ^e,f^	9.97 ^b,c,d^	147.36 ^d^	59.66 ^b^
Kawgum	0.89 ^g^	1.95 ^c,d^	1.34 ^d,e^	10.07 ^b,c,d^	124.17 ^e^	57.44 ^b,c^
Glam Luem Phua	1.54 ^a^	2.01 ^b,c,d^	1.29 ^e,f^	10.51 ^b^	182.74 ^b^	68.63 ^a^
Medmakham	1.29 ^d,e^	2.29 ^b^	2.13 ^a^	8.34 ^d,e^	205.46 ^a^	65.76 ^a^
Deang Sakonnakhon	1.35 ^c,d,e^	2.28 ^b^	1.44 ^d^	8.48 ^d,e^	144.28 ^d^	53.10 ^d^
Sang Yod	1.45 ^a,b,c^	2.78 ^a^	1.79 ^b^	10.03 ^b,c,d^	121.47 ^e^	54.84 ^c,d^
Kawniewdeang	1.40 ^b,c,d^	2.25 ^b,c^	1.30 ^e,f^	9.83 ^b,c,d^	144.47 ^d^	60.25 ^b^
Mali Deang	1.26 ^e^	2.18 ^b,c,d^	1.57 ^c^	10.30 ^b,c^	168.68 ^c^	52.62 ^d^
KDML105	1.27 ^d,e^	2.30 ^b^	1.20 ^f^	7.87 ^e^	10.76 ^g^	6.63 ^f^
RD6	1.06 ^f^	1.93 ^d^	1.06 ^g^	8.70 ^c,d,e^	8.07 ^g^	9.52 ^f^

Means in the same column in the same group of variables (type of rice and variety) followed by the same letter are not significantly different at *p* < 0.05 by DMRT. ^1^ Percentage of control absorbance.

**Table 3 foods-13-00393-t003:** Means for ash content, lipid content and fiber content of 11 genotypes from brown rice and parboiled rice.

Variety	Ash Content (%)			Lipid Content (%)			Fiber Content (%)		
	Brown	Parboiled	*t*-Test	Brown	Parboiled	*t*-Test	Brown	Parboiled	*t*-Test
Glam Feang	1.30 ^c,d^	1.37 ^b^	ns	1.83 ^c,d^	2.35 ^b,c^	*	1.52 ^d^	1.20 ^d^	**
Glam Tonkeaw	1.60 ^b^	1.39 ^b^	**	2.19 ^b,c,d^	2.32 ^b,c^	ns	1.22 ^f^	1.31 ^c,d^	ns
Kawgum	1.28 ^c,d,e^	0.50 ^d^	**	1.62 ^d,e^	2.28 ^b,c^	ns	1.50 ^d,e^	1.18 ^d^	**
Glam Luem Phua	1.40 ^c^	1.68 ^a^	*	1.77 ^c,d^	2.25 ^c^	ns	1.40 ^d,e^	1.19 ^d^	**
Medmakham	1.17 ^e^	1.42 ^b^	**	2.15 ^b,c,d^	2.43 ^b,c^	ns	2.30 ^a^	1.97 ^a^	**
Deang Sakonnakhon	1.18 ^d,e^	1.52 ^a,b^	**	2.28 ^b,c^	2.28 ^b,c^	ns	1.38 ^e^	1.51 ^b,c^	ns
Sang Yod	1.84 ^a^	1.07 ^c^	**	3.34 ^a^	2.22 ^c^	**	2.00 ^b^	1.59 ^b^	**
Kawniewdeang	1.33 ^c^	1.48 ^a,b^	ns	2.35 ^b,c^	2.15 ^c^	ns	1.71 ^c^	0.88 ^e^	**
Mali Deang	1.17 ^e^	1.35 _b_	**	1.81 ^c,d^	2.55 ^a,b^	**	1.16 ^f^	1.99 ^a^	**
KDML105	1.18 ^de^	1.37 ^b^	**	2.45 ^b^	2.15 ^c^	*	1.12 ^f^	1.28 ^d^	ns
RD6	0.74 ^f^	1.39 ^b^	**	1.13 ^e^	2.73 ^a^	**	0.78 ^g^	1.34 ^c,d^	**
Average	1.29	1.32		2.08	2.33		1.46	1.40	

Means in the same column followed by the same letter are not significantly different at *p* < 0.05 by DMRT. ns, *, ** = non-significant and significant at 0.05 and 0.01 probability levels, respectively.

**Table 4 foods-13-00393-t004:** Means for protein content, total phenolic content and DPPH of 11 rice genotypes from brown rice and parboiled rice.

Variety	Protein Content (%)			Total Phenolic Content (mg/100 g Seed)			DPPH (%) ^1^		
	Brown	Parboiled	*t*-Test	Brown	Parboiled	*t*-Test	Brown	Parboiled	*t*-Test
Glam Feang	12.72	12.49 ^a^	ns	114.68 ^h^	21.56 ^d^	**	45.41 ^c^	19.13 ^d^	**
Glam Tonkeaw	10.17	9.77 ^b,c^	ns	237.58 ^e^	57.13 ^b^	**	83.59 ^b^	35.74 ^c^	**
Kawgum	9.68	10.47 ^b^	*	194.75 ^g^	53.59 ^b^	**	84.02 ^b^	30.90 ^c^	**
Glam Luem Phua	10.28	10.75 ^b^	ns	286.18 ^c^	79.29 ^a^	**	85.24 ^a,b^	52.01 ^a^	**
Medmakham	8.35	8.33 ^d,e^	ns	358.25 ^a^	52.66 ^b^	**	87.43 ^a^	44.06 ^b^	**
Deang Sakonnakhon	8.52	8.44 ^c.d^	ns	267.56 ^d^	21.00 ^d^	**	87.17 ^a^	19.04 ^d^	**
Sang Yod	9.34	10.72 ^b^	ns	221.19 ^f^	21.75 ^d^	**	87.30 ^a^	22.38 ^d^	**
Kawniewdeang	9.89	9.77 ^b,c^	ns	244.09 ^e^	44.84 ^c^	**	87.31 ^a^	33.17 ^c^	**
Mali Deang	11.25	9.35 ^b,c,d^	ns	313.74 ^b^	23.61 ^d^	**	87.47 ^a^	17.75 ^d^	**
KDML105	8.77	6.97 ^e^	*	15.42 ^i^	6.11 ^e^	ns	6.95 ^e^	6.32 ^e^	ns
RD6	8.92	8.48 ^c,d^	ns	8.90 ^i^	7.19 ^e^	ns	11.05 ^d^	7.99 ^e^	**
Average	9.81	9.59		205.67	35.34		68.45	26.23	

Means in the same column followed by the same letter are not significantly different at *p* < 0.05 by DMRT. ns, *, ** = non-significant and significant at 0.05 and 0.01 probability levels, respectively. ^1^ Percentage of control absorbance.

**Table 5 foods-13-00393-t005:** Correlation coefficients (r) among moisture content, fiber, ash, lipid, protein, phenolic acid and DPPH of 11 rice varieties.

	Moisture	Fiber	Ash	Lipid	Protein	Phenolic
Fiber	−0.1199					
Ash	0.2044	0.2611 *				
Lipid	−0.0994	0.3932 *	0.4357 **			
Protein	0.0605	−0.1601	−0.0137	−0.0775		
Phenolic	0.0343	0.3409 **	0.1166	−0.0926	0.0633	
DPPH	0.0341	0.3391 *	0.2308	0.4357 **	0.1270	0.9470 **

*, ** = significant at 0.05 and 0.01 probability levels, respectively.

## Data Availability

Data are contained within the article.

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
