# Peer review of "Effects of Parboiling on Chemical Properties, Phenolic Content and Antioxidant Capacity in Colored Landrace Rice"

_foods, 2024, doi:10.3390/foods13030393_

Round 1

Reviewer 1 Report

Comments and Suggestions for Authors

This manuscript is described as “Effects of parboiling on chemical properties, phenolic content, and antioxidant capacity in colored landrace rice." This study evaluated changes in chemical content, total phenol content, and antioxidant capacity of landrace rice genotypes under parboiled conditions, and genotypes suitable for parboiled rice production were identified. This study contributes to understanding how parboiling affects rice composition and nutritional value. Information on varieties with high phytochemical and antioxidant content can be helpful for direct applications in rice breeding. Here are some suggestions for the authors to improve the manuscript.

1. In making parboiled rice, it is necessary to explain the reason or purpose of germinating for 48 hours after soaking it for 24 hours, as there are several methods for making parboiled rice.

2. In this study, it is believed that the changes in nutrients in parboiled rice are mainly due to germination. Many papers have been published on the biochemical changes in germinated brown rice. It is recommended that this germination effect be addressed in this manuscript's introduction and discussion sections. 

3. Line 55: Is the expression “meat” appropriate in this sentence?

4. Line 90: Abbreviations require explanation.

5. Table 2 and result 3.2: Please make it clear that the values for each variety in Table 2 represent the average values for brown rice and parboiled rice.

Comments on the Quality of English Language

This manuscript requires grammatical improvements, such as adding commas before and, using plural forms, and adjusting tense.

Author Response

Dear Editor and reviewers,

The manuscript is revised according to the comments and suggestions of the reviewers. Yellow highlights are indicated where the manuscript is changed. The new lines are also provided.

Reviewer 1

Comments and Suggestions for Authors

This manuscript is described as “Effects of parboiling on chemical properties, phenolic content, and antioxidant capacity in colored landrace rice." This study evaluated changes in chemical content, total phenol content, and antioxidant capacity of landrace rice genotypes under parboiled conditions, and genotypes suitable for parboiled rice production were identified. This study contributes to understanding how parboiling affects rice composition and nutritional value. Information on varieties with high phytochemical and antioxidant content can be helpful for direct applications in rice breeding. Here are some suggestions for the authors to improve the manuscript.

  1. In making parboiled rice, it is necessary to explain the reason or purpose of germinating for 48 hours after soaking it for 24 hours, as there are several methods for making parboiled rice.

Although Thailand is a leading rice exporter, parboiled rice is not sold in the local market. Parboiled rice is mainly for export. There are several methods of parboiling in different countries. The method used in this study is used in parboiled rice industry in Thailand. The major objectives of parboiling are to: increase the total and head yield of the paddy, prevent the loss of nutrients during milling, salvage wet or damaged paddy, and prepare the rice according to the requirements of consumers.

There are several changes after parboiling. The changes are mainly beneficial for conserving rice quality for long term storage although there are some unfavorable effects. Extended incubation time to 48 hours can lead to an increase in useful phytochemicals, and, therefore, we selected 48 hours of incubation time for this study.

  1. In this study, it is believed that the changes in nutrients in parboiled rice are mainly due to germination. Many papers have been published on the biochemical changes in germinated brown rice. It is recommended that this germination effect be addressed in this manuscript's introduction and discussion sections.

The Introduction section is revised (line 56-59), and more discussion is included in the Discussion section.

  1. Line 55: Is the expression “meat” appropriate in this sentence?

Meat to grain (new line 56)

  1. Line 90: Abbreviations require explanation.

AOAC is defined as the Association of Official Agricultural Chemists in line 95 (new line).

  1. Table 2 and result 3.2: Please make it clear that the values for each variety in Table 2 represent the average values for brown rice and parboiled rice.

 The value in Table 2 was present as the main effect of this study, type of rice and rice genotypes. So, we changed the title of the table to “Table 2. Means for ash content (%), lipid content (%), fiber content (%), protein content (%), total phenolic content (mg/100 g seed) and DPPH (%) of brown rice, parboiled rice and 11 rice varieties”.

We rewrite the 3.2 title from “Mean across two types of rice” to “Means of brown rice, parboiled rice and 11 rice varieties”

Reviewer 2 Report

Comments and Suggestions for Authors

The manuscript is well written, clear and it provides new scientific evidence in the field. However, there are few aspects that demand improving. Below I list remarks that shall be considered by the author: 

Abstract

The authors are kindly asked to include the names of the cultivars and genotypes studied in the abstract. In addition, an explanation, or a definition of what is understood as “brown rice” is needed. After each cultivar name an abbreviation “cv.” shall be written – this applies to the entire manuscript and not the abstract only.

Key words – a key word „pigmented rice” is mentioned; however, this term cannot be found anywhere in the manuscript. Brown rice is used instead. Please choose the right key words and/or explain or define “ pigmented rice”.

Introduction

The introduction section lacks a hypothesis. Please add.

Materials and Methods

An explanation or a definition of what is understood as “brown rice” is needed.

The authors are studying the following cultivars:  five black landraces, four red landraces and two white commercial varieties. The authors are encouraged to include a short description of each cultivar or group of cultivars.  

Data analysis

The factors in this study were rice type and cultivar. The authors shall explain what is meant by „rice type”. Clear description of this shall be added to the Materials and Methods section.

Discussion

The authors claim that soaking time dependent changes in the ash content were found in parboiled rice. “Long time soaking reduced ash content whereas short time soaking increased ash content.” It is necessary to specify how long short and long soaking time is exactly. This study did not find a correlation between parboiling and ash content and that is opposite to the results found by authors of cited literature. The authors are asked to discuss and attempt to explain this discrepancy.

In row 326 the following sentence is featured: „In our study, the range of lipid content was between 1.93% and 2.78” however it is not clear which type of rice this statement refers to. Please clarify.

The authors write that parboiling greatly reduced total phenolic content, however, “contrasting results have been reported in previous studies”. The authors are kindly asked to try to explain and discuss this discrepancy.

Conclusions

The conclusion shall be more clear and a clear statement explaining if the proposed method of parboiling had a positive influence on quality shall be included. Additionally. Kindly include a clear statement informing which rice cultivars result in best quality rice after parboilding.

Author Response

Dear reviewer,

The manuscript is revised according to the comments and suggestions of the reviewers. Yellow highlights are indicated where the manuscript is changed. The new lines are also provided.

Reviewer 3 Report

Comments and Suggestions for Authors

Dear Authors,

I revised the paper “Effects of parboiling on chemical properties, phenolic content and antioxidant capacity in colored landrace rice” submitted to Foods. It matches the aim and scope of the Journal, I suggest major revisions.

Please, see comments below.

Abstract

Please, mind that the abstract exceeds 250 words (please, see Instructions for Authors).

Introduction

It should be reformulated. In my opinion, lines 35-50 are not necessary. Moreover, the novelty of the study should be stressed. Please, mind that the effects of parboiling on nutritional properties of rice are well-known.

Materials and methods

Line 90: please, specify method number.

Line 101: please, specify method number.

Please, report how many replicates were performed for each parameter.

In addition to the above-mentioned comments, the study has some critical points:

1.       It seems that parboiling has different effects on rice landraces under investigation. Does this imply that some varieties are not suitable for parboiling?

2.       Can parboiling be tailored to each rice variety so to maximize the nutrient content? 

Author Response

(The authors gave the same response as above.)

Round 2

Reviewer 3 Report

Comments and Suggestions for Authors

Dear Authors,

thank for your replies. The amended paper is now suitable for publication.